# Reaching Natural Growth: The Significance of Light and Temperature Fluctuations in Plant Performance in Indoor Growth Facilities

**DOI:** 10.3390/plants9101312

**Published:** 2020-10-05

**Authors:** Camilo Chiang, Daniel Bånkestad, Günter Hoch

**Affiliations:** 1Department of Environmental Sciences—Botany, University of Basel, 4056 Basel, Switzerland; guenter.hoch@unibas.ch; 2Department of Research and Development, Heliospectra, 414 58 Gothenburg, Sweden; daniel.bankestad@heliospectra.com

**Keywords:** dynamic light, dynamic temperature, natural growth, controlled environment

## Abstract

Recommendations for near-natural plant growth under indoor conditions have been described without considering environmental fluctuations, which might have important consequences for researchers and plant producers when comparing results from indoor facilities with natural ecosystems or production. Previous authors proposed that differences in temperature, light quantity, and the lack of their variation are sources of deviations between indoor and outdoor experiments. Here, we investigated the effect of fluctuating light, temperature, and humidity in an indoor environment on plant performance. Seven plant species from different functional plant types were grown outdoors during summer and spring. The same species were then grown in indoor growth chambers under different scenarios of climate complexity in terms of fluctuations of temperature, air humidity, and light: (1) fixed night and day conditions, (2) daily sinusoidal changes, and (3) variable conditions tracking the climate records from the field trials. In each scenario, the average of the environmental variables was the same as in the respective field trial. Productivity-, gas exchange-, and leaf pigment-traits were measured in all plants at the end of the experiments. The plant trait responses were highly dependent on species and treatment, but general trends were observed. The variable condition yielded lower biomass compared to the fixed and sinusoidal conditions, together with a higher specific leaf area and increased chlorophyll concentrations. A principal component analysis (PCA) across all plant traits in response to climatic conditions suggested that at least a sinusoidal fluctuation is recommended for a more natural-like plant performance in indoor growth facilities. However, prevailing significant differences for several traits between field- and indoor-grown plants even under variable climates indicate that additional factors other than those controllable in standard phytotrons (e.g., wind speed and direction, leaf and soil temperature) can still significantly bias plant performance in indoor facilities.

## 1. Introduction

From a scientific and commercial point of view, natural-like (i.e., similar to outdoor conditions) growth and performance of plants are often desired in indoor facilities [1]. Although it is well known that several environmental interactions may affect plant phenology, and therefore the outcome of experiments, it is common practice to apply static environmental conditions in indoor experiments. Fixed day and night-time conditions may represent an oversimplification of natural conditions and may lead to results significantly deviating from outdoor-grown plants [2]. Two of the most important environmental factors that affect plant growth are light and temperature. Instantaneous and daily fluctuations of temperature and light can affect plant performance in both positive and negative ways [3,4]. Myster and Moe [4] reviewed the effect of the difference between day and night temperatures, where a positive difference between day and night temperatures enhanced plant height, chlorophyll content, and leaf orientation (more upright position), mainly due to an increase in cellular elongation. Since cell metabolism is not linearly related to temperature, an increase in temperature may induce a stronger effect than a decrease in temperature of the same magnitude. Rapid changes in temperature can adapt the plants to less favorable conditions, i.e., “hardening” [1]. Daily and instantaneous changes in light have also been studied in detail previously. From these studies it is known that changes in light during the day may induce lower biomass but also higher maximum photosynthesis (A_max_), especially per unit of leaf mass [5]. Fast fluctuations in light intensity have been shown to reduce photosynthesis and biomass in the long term [6], partly related to increases in radical oxygen species (ROS) and interactions with other environmental factors. Under increasing light, higher leaf temperatures can lead to stomatal closure and limited photosynthesis [7]. Meanwhile, under reductions of light, light use efficiency could be slowed down due to the relaxation of energy dissipation [8]. Interestingly, fluctuating light has in some cases been shown to promote several photosynthesis-related parameters, especially in partially shaded leaves [9]. Several studies have measured the effect of light or temperature variations on plant performance under semi-controlled and controlled conditions, but simultaneous comparisons with outdoor grown plants are scarce.

Having static climatic conditions in indoor plant research is often practical and convenient, but the generated results may not extrapolate well to natural conditions as the important factor of environmental fluctuations is missing [1,2]. Hence, contradictory results have been found when similar treatments have been applied in indoor and outdoor experiments, e.g., Strømme et al. [10] found that for *Populus tremula* grown under outside conditions, an increase in temperature promoted bud break, but other authors claimed the opposite for greenhouse-grown *P. tremula* trees [11,12]. These results showcase the difficulties to translate indoor results to real-world conditions, and when trials have been conducted to replicate outdoor growth in indoor facilities, low correlations have been found [13,14]. Poorter et al. [2] suggested multiple reasons why this may occur, where the main differences may come from lower light quantities, higher plant density, and shorter durations of indoor compared to outdoor experiments. Other sources of variation have been pointed out, including the age of the plants, leaf temperature, soil temperature, soil microorganisms, lack of UV light, and the light quality in indoor experiments [2]. When the effect of light quality was studied under constant day and night conditions of temperature, light quality was shown to affect plant morphology [15,16]. 

The aim of this study was to compare and quantify the effects of fluctuating environmental conditions on several important plant traits (productivity, gas-exchange, and leaf pigmentation), aiming to reach a close to natural plant performance under indoor growth conditions. Seven plant species from different functional groups (see material and methods for details) were included in two 35-day field trials where the in situ climate was recorded and plant traits were measured at the end of the trials. The same plant species were then grown in indoor experiments in phytotrons under three different levels of complexity of light, temperature, and air humidity fluctuations: fixed, sinusoidal, and real-time tracking that had the same average as the outdoor conditions. We hypothesized that applying steady, average climatic conditions will lead to plant growth that deviates most from natural growth, while the application of real fluctuations of temperature, humidity, and light will produce plants that show similar performance to field-grown plants.

## 2. Results

### 2.1. Environmental Conditions

After tracking the environmental conditions of the outdoor trial (Figure 1), three different environmental condition complexities were used as treatments, with the average conditions shown in Table 1.

### 2.2. Plant Growth and Morphology

In both runs (summer and spring) there was an interactive effect between treatment and species on plant height (Table 2). On average across all species, the sinusoidal climate change was the only treatment that did not result in significantly different plant heights compared to the field trial (Figure 2A,B). Fixed and variable conditions, induced, on average, lower and higher heights than the outdoor treatment in the summer and spring runs, respectively (Figure 2A,B). Although there was a considerable spread of the species around the average values, it is interesting to note that under summer conditions, most plants showed lower height growth compared with the field trial, while it was the opposite for the spring conditions (Figure 2A,B). 

An interaction between treatment and species was also found for each trial in terms of total biomass (Table 2), where some species (especially *Alnus, Melissa*, and *Raphanus*) showed large deviations from the outdoor results in one or both phytotron runs (Figure 2C,D). However, when averaged across species, lower total biomass was reached under the variable conditions compared with the fixed condition independent of the run (Figure 2C,D). In addition, the species mean total biomass did not differ significantly from the outdoor trials under the variable climate, while it was significantly increased under the fixed climate in both runs, and in the sinusoidal treatment for the summer run. The biomass of individual organs largely followed the total biomass trends for all treatments and in both runs (Data not shown; available as Appendix A).

Similar to all other growth traits, there was also a significant interaction between treatment and species on the root to shoot ratio (r:s; Table 2). In the summer run, higher ratios were obtained in the three indoor treatments compared with the outdoor treatment for almost all species (Figure 2E,F). This was not the case in the spring run, where lower values were obtained with the exception of *Raphanus*. *Raphanus* had higher r:s ratios under fixed and sinusoidal conditions compared with outdoor or variable conditions. On average across species, only the variable treatment yielded significantly lower values (i.e., higher allocation to shoot biomass) than the outdoor treatment in the spring run. 

In both runs and for most species, the sinusoidal and variable conditions induced a higher specific leaf area (SLA) compared to the field trial, while SLA tended to be lower under the fixed conditions (Figure 2G,H). Although the trend was similar among species, an interaction was found between treatments and species on the SLA as well (Table 2). The effect of the different treatments within trials was greatest for *Lactuca* and *Ocimum*, where the difference in SLA was almost two-fold among treatments. 

### 2.3. Leaf Pigmentation and Leaf Gas-Exchange

Compared with the fixed treatment, higher concentrations of chlorophyll a were reached under variable environmental conditions in all species and both runs. but this effect was stronger in the spring run (Figure 3A,B). An interaction between species and treatment was found in both runs (Table 2). The sinusoidal treatment, on average across species, did not differ from the outdoor treatment, but fixed conditions induced lower concentrations compared with the outdoor treatment (Figure 3A,B) in both runs. Chlorophyll b followed the reactions of chlorophyll a for most of the species (Appendix A). Hence, the chlorophyll a:b ratio was similar among treatments (Figure 3C,D). Nevertheless, there was a significant interaction between species and treatments for the summer, but not for the spring run (Table 2). On average, higher chlorophyll a:b ratios were recorded only under fixed conditions compared with all the other treatments in the summer run (Figure 3C,D).

Chlorophyll fluorescence, measured as *Fv/Fm,* values were around 0.8 in the field trial and all phytotron treatments (Appendix A), indicating the absence of significant stress in all treatments. However, *Fv/Fm* values were higher under sinusoidal conditions compared with the fixed treatments in both runs, and also under the variable treatment in the spring run (Figure 3E,F). A significant interaction between species and treatment was found for *Fv/Fm* (Table 2), as well as for the fluorescence Pi value (data not shown).

In contrast to most growth traits, no interaction was found between the treatment and species on their effect on photosynthesis at 700 μmol m^−2^ s^−1^ PPFD (A_700_) in both runs, and additionally, there was no statistically significant effect of treatment in the spring run when species and treatment were considered fixed variables (Table 2). On average across all species, plants had higher A_700_ values compared to the outdoor treatment under the fixed climatic treatments in both runs and under the variable treatment in the summer run (Figure 4A,B), largely driven by the strong reactions of *Lactuca* and *Ocimum*. The light compensation point of net photosynthesis was not significantly affected by the different treatments in the summer trial (Table 2). The light compensation values from the spring trial showed no interactive effect between treatment and species, but much lower compensation points were reached for all the indoor treatments compared with the outdoor treatment (Figure 4C,D). This trend was strongest for the variable conditions, which induced lower compensation points on average across species than the other two indoor treatments. Only in the summer run was an interactive effect found between treatment and species for the quantum yield of CO_2_ fixation (α) (Table 2). On average across species, α was higher during the summer trial under fixed and variable conditions, compared with the sinusoidal and outdoor treatments, especially driven by *Triticum*. In the spring run, the average α across species was significantly higher under the variable conditions compared with the outdoor treatment, while no significant difference from the outdoor treatment was found for the sinusoidal and the fixed treatment (Figure 4E,F). There was no treatment effect on leaf respiration in the dark in the summer run, but a significant treatment effect and a treatment × species interaction in the spring run, with lower values of respiration, in the fixed and variable treatments (Table 2, Appendix A).

#### Principal Component Analysis (PCA)

A PCA performed separately for each species and the two runs showed clear separations among the treatments in most cases (Figure 5). The sinusoidal and variable treatments were often more clustered whilst the fixed and outdoor treatments were furthest from the mean, although this was not always the case. Interestingly, within a species, the treatment grouping was very different between the summer and the spring runs (Figure 5). Within the PCAs, traits of the same type (biomass, pigments, photosynthesis) tended to point in similar directions (Available as Appendix A), with the exception of light-related parameters, that often pointed along different axes within a species. Independent of the species or run, the effect of the different factors had similar weights, and the first two principal comments explained, on average, 57% of the total variance (standard deviation = 3.5%).

## 3. Discussion

The incorporation of environmental variability has previously been recommended for more natural-like plant growth [1,2,17,18], but is rarely applied in phytotron studies due to practical reasons and technical limitations of the growth chambers. Within this study, we found significant differences in almost all investigated plant traits among the different climatic scenarios applied indoors, but also strong differences with the plants from the outdoor trials. Although there was an overall trend to more similar traits to the outdoor plants in phytotron runs that simulated the real temperature, light, and humidity variations, this was not the case for all traits. Importantly, we also found high species-specificity. Overall, we could show that the type of environmental variation does affect plant morphology and photosynthetic capacity, in line with previous studies [6,7], and with studies that looked at specific metabolic processes [17,18].

### 3.1. Plant Growth and Morphology

Although the effect size of climatic variability on plant height differed between the summer and spring run, in both runs the sinusoidal treatment did not differ from the outdoor control across species. It has been suggested that a larger difference between day- and night-time temperature can stimulate the production of abscisic acid, which may enhance stem elongation and help to respond to changes in the environmental conditions [19,20]. On the other hand, it has also been proposed that daily light fluctuations can induce shorter plants [2]. The results from the present study suggest that in our case, the diurnal temperature amplitude may have had a larger effect on plant height than the presence of fluctuating light.

The reduction in total plant biomass under fluctuating climates was one of the strongest effects in this study and has also been reported in several previous experiments [2,5,18]. In our case, the difference between treatments was more marked in the spring run, in which not only the totally variable conditions but also the sinusoidal treatment did not differ from the outdoor control. This may suggest that the main effects of a fluctuating environment without extreme conditions may lie in the amplitude between the daily minimum and maximum temperatures, as previously mentioned by Annunziata et al. [18]. Annunziata et al. [17] could show for *Arabidopsis thaliana* that there is a lower accumulation of starch during the day under fluctuating environments, which also implies that the plants have less starch available during the night for growth and metabolism [21]. Additionally, lower night temperatures will also reduce the consumption of C reserves because of reduced metabolic and growth activity [22]. Annunziata et al. [18] showed that under variable light and temperature, 90% more biomass could be reached in *A. thaliana* compared with fluctuating light and fixed temperature, corroborating the relevance of joint environmental fluctuations for plant growth. 

The observed differences in root: shoot ratios between phytotron and outdoor experiments might result from deviating pot soil temperatures. Although pot temperature was not monitored or controlled in our experiment, it can be assumed that under natural sunlight, the soil is likely to warm up more compared with under the light emitting diodes (LED) lights in the phytotrons. Differences between air- and soil-temperature may play an important role in the allocation of reserves and new biomass production [23,24]. In our experiment, the differences in root: shoot ratios between indoor and outdoor conditions showed the reversed pattern to the stem biomass, where the root: shoot ratio was higher in all phytotron treatments in the summer run, but lower in the spring run compared with the control (Figure 2). Although the 35-day mean temperatures were similar between the summer and the spring runs, the spring run had considerably lower temperatures in the first couple of days (Figure 1). The potentially stronger effect of the above-mentioned outdoor soil warming during the cooler spring days might have led to an early divergence in the biomass allocation between outdoor and phytotron plants that affected the whole experiment differently than in the summer run. Although several reports are available about the effects of light and temperature fluctuations on plant growth and physiology [5,17,18,25], none of them focused on biomass allocation changes.

Plants that were treated with fluctuating climates produced thinner leaves with significantly higher SLA values compared to plants treated with the fixed conditions. This result is in line with a recent indoor study which found that the SLA in *Arabidopsis thaliana* was up to 25% higher with a thinner spongy mesophyll layer under fluctuation levels of light and constant temperatures, compared with a fixed light treatment lacking light fluctuations [5]. Under natural conditions, greater climatic fluctuations, especially the presence of cold temperatures and high solar radiation, are generally associated with the production of hardened leaves and smaller SLA values. In particular, sunlight-adapted leaves tend to be thicker compared with shade-adapted leaves, as a result of the compromise between the increase in the chloroplast surface area for CO_2_ dissolution, due to the low affinity of Rubisco for CO_2_, and the production costs of thicker leaves [26]. The fact that we found increasing SLA values with increasing climatic variation suggests that neither the applied minimum and maximum temperatures nor the applied light peaks were extreme enough to induce thicker or denser leaves. Finally, our study as well as the above-mentioned experiments [5,17,18] were performed without UV light, which is known to reduce stem elongation and SLA [27].

### 3.2. Leaf Pigmentation and Photosynthesis

To our knowledge, an increase in chlorophyll under fluctuating environmental conditions has not been reported so far. Higher light intensities have been related with lower Chl a and b concentrations in leaves (on a dry matter basis), but temperature and water availability have been identified as the most important factors for the synthesis of chlorophyll [28]. Erwin and Heins [29] showed a positive correlation between the diurnal difference between day and night-time temperatures and total chlorophyll leaf concentrations in *Dendranthema* and *Chrysanthemum*. Similarly, chlorophyll concentrations were significantly higher in the variable compared to the fixed climate treatments in the current study. Similar to Vialet-Chabrand et al. [5], who reported higher chlorophyll a:b ratios in *Arabidopsis* under fixed vs. variable light conditions (4.27 vs. 3.72 on average, respectively), we did find significant changes in chlorophyll a:b, especially between the fixed and variable conditions (Table 2).

It is broadly assumed that fluctuating light can increase the photosynthetic capacity of plants [5,9,25,30]. In line with this notion, we found increased *Fv/Fm* values in the sinusoidal and variable compared to the fixed treatments in our experiments. Vialet-Chabrand et al. [5] also reported higher *Fv/Fm* values under variable light conditions, which might be related to a higher Photosystem II (PSII) capacity. In contrast, several authors have shown lower *Fv/Fm* values under varying environmental conditions. Yamori [7] explained such reductions mainly by a photoinhibition of PSI under fluctuating light, while other abiotic stressors were linked to a photoinhibition of PSII. Additionally, it has been proposed that fluctuations in temperature over the day and night may have a reparative effect in both photosystems, due to a better circadian clock adjustment [18], resulting in higher *Fv/Fm* values under fluctuating environments.

Similar to the photosynthetic capacity, previous studies have indicated that Amax may be enhanced in plants that grow in fluctuating light environments. Vialet–Chabrand et al. [5] highlighted that this effect becomes especially evident if photosynthesis values are calculated per leaf mass instead of leaf area. Mathews et al. [25] demonstrated that the time of the day when the measurements are conducted have an important impact on the effect size of light fluctuation on photosynthesis. Especially in the morning, fluctuating light treatments tended to produce higher A_max_ rates than at midday, compared with both sinusoidal and square light treatments [25]. Poorter et al. [2] reported generally lower values in indoor vs. outdoor experiments, especially in woody species, probably also driven by the often higher SLA of indoor vs. outdoor plants.

In our study, there were no strong differences in the light compensation point as well as CO_2_ yield of photosynthesis among the phytotron runs, but significantly lower compensation points in the phytotrons compared to field-grown plants in the spring run, and higher CO_2_ yields in indoor plants under simulated summer and spring conditions. The latter might be a consequence of the different light spectral composition of the used LED lamps in the phytotrons (with a higher proportion of blue and red light compared to the sun spectrum). Vialet-Chabrand et al. [25] did not find any difference in the photosynthetic light compensation point when comparing fixed and fluctuating light treatments in *Arabidopsis thaliana* under constant temperature, whereas light intensity was shown to play a more important role.

### 3.3. Principal Component Analysis

When all the different variables were analysed through the PCA (Figure 5 and Appendix A), several species tend to have either the fixed or the variable treatment separated furthest from the rest of the treatments. Even though the average environmental variables were all the same for the different treatments within each run, the PCA clearly separated each treatment in all species, suggesting that some treatments give more “natural-like growth”, the closer the trial is to the field trial. In our case, several species under the sinusoidal and the variable treatments were grouped closer to the outdoor-grown plants than the plants grown under fixed conditions with only a few exceptions. Annunziata et al. [17] compared a fixed and a sinusoidal light treatment against natural light to evaluate metabolism effects in *Arabidopsis* under stable temperature and found similar variances between light treatments and indoor and outdoor plants based the first two principal components (37–34% and 23–16% for the first and second component, respectively). However, different from the current study, Annunziata et al. [17] found a much clearer separation between the outdoor treatment and the indoor treatments, but lower differences between the indoor treatments. Posteriori, Annunziata et al. [18] demonstrated that after removing the temperature fluctuation under either fluctuating or fixed light, there was an increase in scattering of the PCA values between treatments, hypothesizing that fluctuating temperature had a stronger effect than light fluctuation on the plants’ metabolism in their experimental scenario.

## 4. Materials and Methods 

### 4.1. Plant Material and Pre-Growing Conditions

In total, we investigated seven species from different functional plant types: trees represented by black alder (*Alnus glutinosa* L., provenance HG4, Zurich, Switzerland, Swiss federal institute for forest, snow and landscape research (WSL), Switzerland) and scotch elm (*Ulmus glabra* HUDS., provenance Merenschwand, Aargau, Switzerland, WSL, Switzerland), herbs represented by basil (*Ocimum basilicum* L. var Adriana, Wys Samen Pflanzen, Switzerland), lettuce (*Latuca sativa* L., Wys samen pflanzen, Switzerland), melissa (*Melissa officinalis* L., Wys Samen Pflanzen, Switzerland) and radish (*Raphanus raphanistrum* L. subsp. sativus var. Marabelle, Wys Samen Pflanzen, Switzerland) and finally grasses represented by winter wheat (*Triticum aestivum* L., Sativa, Switzerland). Throughout the text, the species are always referred to only by their scientific genus name for clarity. Due to the different germination speeds, the timing of sowing was different for the species as follows: Seeds of black alder and scotch elm were sown in 20 cm × 40 cm × 2 cm trays with a commercial substrate (pH 5.8, 250 mg L^−1^ N, 180 P_2_O_5_ mg L^−1^, K_2_O 480 mg L^−1^, Ökohum, Herrenhof, Switzerland) 43 days before the start of the experiments and left to germinate under 190 μmols m^−2^ s^−1^ of photosynthetic photon flux density (PPFD) with a red to far-red ratio (R:FR) of 5.1 for 23 days. Twenty days before the start of the experiment, the light was increased to 240 μmols m^−2^ s^−1^ PPFD, with a R: FR of 5.1, to acclimate the plants to higher light levels. Thirteen days before the start of the experiment Melissa, and six days before the start of the experiments the rest of the species were sown in the same type of trays and under the same environmental conditions with the exception of *Triticum*, which was sown directly in round 2 L pots with a density of 15 seeds per pot. During the pre-growing period the seedlings were exposed to 25/15 °C and 50/83 % relative humidity (RH) during the day and night, respectively, with a daylength of 10 hours and one-hour light/temperature/humidity ramping pre and post daytime.

At the start of the different treatments, all species except *Triticum* were transplanted into 2 L cylindrical pots of 13.5 cm diameter (Pöppelmann, Lohne, Germany), with a single individual per pot. The pots were filled with the same substrate used in the germination trays. During the experiments, all plants were watered daily at the beginning of the day. At the beginning of the experiments, each pot was fertilized with 4 g of a slow-release fertilizer (Osmocote exact standard 3–4 months, Scotts, Marysville, OH, USA) containing 16% total N, 9% P_2_O_5_, 12% K_2_O, and 2.5% MgO.

### 4.2. Outdoor Trial and Environmental Conditions

Nine plants of each species, pre-grown under the conditions given above, were grown under outdoor conditions for a period of 35 days during summer (4 August 2017–7 September 2017) and spring (15 May 2018–18 June 2018) in an open site at the botanical garden of the University of Basel, Basel, Switzerland. Both trials were used as control treatments for two separate rounds of phytotron experiments. All pots were placed on a metal grid (the same grid was also used in the indoor runs) with a density of 30 pots per m^2^, and all plants were watered daily, to avoid any influence of soil water limitation. Temperature, relative humidity, precipitation, wind speed/direction, and PPFD (400–700 nm) were recorded every 5 min using a weather station (Vantage pro2, Davis, Haywards, CA, USA). In addition, sunlight spectra in the waveband 350–800 nm were recorded every minute using a spectrometer (STS, Ocean Insight, FL, USA) that was equipped with an optical fiber and a cosine corrector (180° field-of-view; CC-3-UV-S, Ocean Insight) placed next to the weather station’s PPFD sensor facing upwards. The spectrometer was connected to a Raspberry Pi 2 computer for automatic sampling, integration time adjustments, and data storage. Posteriori, the spectra were used to calculate photon flux densities within specific wavebands: PPFD (400–700 nm), blue light (400–500 nm), green light (500–600 nm), red light (600–700 nm) and the R:FR ratio (655–665 nm and 725–735 nm [31]). The measurements of light were corroborated through the correlation between the data from the weather station and the PPFD calculated from the spectrum. 

### 4.3. Experimental Phytotron Runs

In two different runs corresponding to summer and spring conditions, three different environmental treatments were applied for 35 days in closed walk-in chambers (phytotrons). The plant species, replication, and pot density were the same as in the respective field trials (see above). Each phytotron (195 × 130 × 200 cm, L × W × H) was equipped with 18 120-cmlong LED panels consisting of a mixture of individually dimmable B, G, R, and FR LEDs per panel with a maximum PPFD of 1200 µmol m^−2^ s^−1^ (DHL-Licht—Prototype, Hangover, Germany) measured 1 m from the light source. The LED lighting system of each chamber was mounted on movable ceilings, the which height could be adjusted by the phytotron control software to alter the distance to the plants, thereby allowing for precise adjustments of the effective radiation strength at canopy height. The three climatic treatments were as follow: (1) Fixed: constant day and night conditions resembling the average day and night-time climate from the 35-day field trial, (2) Sinusoidal: a sinusoidal, average diurnal climate based on the average of five minute recordings from the 35-day field trial between days resulting in a periodic function, and (3) Variable: an exact replication (setpoints every 5 min) of the recorded temperature, humidity, and PFFD from the 35-day field trial (Figure 1 and Appendix A). Due to low germination of *Alnus*, this species was not included in the phytotron experiments under sinusoidal spring conditions. In each treatment, the environmental conditions resulted in the same average values as in the respective field trials across the 35 days (Table 1). The used light spectra in the phytotrons (Figure 2) corresponded to a spectral composition that gives more natural plant growth as derived from a previous experiment (Chiang et al., unpublished). The light intensity was regulated through changes in electrical intensity and roof height, maintaining similar spectra. For moments where this was not possible in the variable treatment, higher amounts of B and R light were applied, keeping the same previously used B:R ratio. The R:FR ratio was kept at 1.8 for all treatments in the phytotron runs due technical limitations. No UV light was applied in the phytotrons, and the airflow (average value of 0.295 m s^−1^) in the chamber came from below, ensuring a uniform temperature and humidity distribution within the chambers.

### 4.4. Plant Growth and Morphology

The height of the nine plants was measured after 35 days of exposure to the different treatments, as total height from the substrate to the apical meristem. In the case of flowering or plants without a clear steam, the extended leaf length was recorded as the height, with the exception of *Lactuca* where height was not recorded. Two fully grown leaves from the top three leaves were taken from each plant to measure surface area (LI-3100, Licor, Licoln, NE, USA) and dry weight to calculate specific leaf area (SLA). For each plant (*n* = 9 per species and treatment), dry weight (DW) was measured separately for roots, stems, and leaves after 10 days drying at 80 °C in a drying oven (UF 260, Memmert, Schwabach, Germany). Due to the lack of a clearly identifiable stem, only total aboveground and root biomass was determined for *Lactuca, Melissa* and *Triticum*. All reported organ masses and the below-to-above biomass ratio (root to shoot ratio; r:s) refer to dry biomasses.

### 4.5. Chlorophyll Fluorescence and Leaf Pigment Content

The night before the end of the experiment, fast chlorophyll fluorescence was measured on one of the top four leaves of four randomly chosen plants of each species and treatment by using a continuous excitation fluorometer (Pocket PEA, Hansatech Instruments Ltd., Nordfolk, UK). The plants were dark adapted for at least 20 min (night measurements), and *Fv/Fm* and Pi absolute [32] were recorded.

At the end of the experiment, two discs of 1.13 cm^2^ from two of the top four fully developed leaves were punched and collected in a 1.5 mL Eppendorf tube together with 4–6 glass beads of 0.1 mm diameter for later chlorophyll and carotenoid analyses in four different plants per species. The tubes were rapidly frozen in liquid nitrogen and then stored at −80 °C until analysis. On the day of pigment measurement, the tubes were agitated using a mixing device (Silamat S6, Ivoclar Vivadent, Schaan, Liechtenstein) during two rounds of 10 s to triturate the tissue. Then, 0.7 mL of acetone was added to each tube, agitated again for 10 s, and then centrifuged at 13,000 rpm at 4 °C for 2 min; 0.25 mL of the supernatant was taken and dissolved in 0.75 mL of acetone. The spectra of the extracts were measured using a spectrometer (Ultrospec 2100 pro, Biochrom, Holliston, MA, USA). Chlorophyll a, b, the chlorophyll a to b ratio, and total carotenoid concentrations were calculated from the spectrum using the absorption values at 470, 646, and 663 nm as described by Wellburn [33] and expressed as mg per g of dry biomass using the average SLA of each species and treatment.

### 4.6. Photosynthesis

Six days before the end of the experiment, three light-response curves of net CO_2_ leaf-exchange were measured on one of the top three leaves of three randomly chosen plants per species using a portable photosynthesis system (LI-COR 6800, Licor, Lincoln, NE, USA). The light reaction-curves were measured under the applied light spectra in the phytotrons (Appendix A) using a clear top leaf chamber. Due to the lower maximum irradiance in the phytotron at the same spectra, the light curves with growing light were measured only up to a maximum radiation of 700 µmol m^−2^ s^−1^ of PPFD (700, 480, 380, 200, 100, 60, 30, 20, 17, 15, and 0 µmol m^−2^ s^−1^ of PPFD) to maintain the spectral quality. All leaf CO_2_-exchange measurements were conducted at 400 µmol CO_2_, 60% relative air humidity, and 20 °C leaf temperature, with 60 to 120 s as the threshold for stability after each light change. Stability of readings was assumed when the difference of the slopes between infra red gas analysers (IRGAs) was smaller than 0.5 and 1 µmol m^−2^ s^−1^ for CO_2_ and H_2_O, respectively. 

For each curve, 12 different light models where fitted, including a model for photo inhibition [34,35], and the model with the lowest sum of squares was selected in each case. From the selected model, four different parameters were calculated: maximum photosynthesis within the range of measured light (A_700_), quantum yield for CO_2_ fixation (α) as the slope of the curve between 0 and 100 µmol m^−2^ s^−1^ of PPFD, dark respiration (DR), and the light compensation point (CP).

### 4.7. Statistical Analysis

To evaluate the effect of the different treatments, two-way analysis of variance (ANOVA) was performed for all the studied variables for each season, considering the species and different treatments as fixed factors (Table 2). The data were checked for normal distribution, independence, and homogeneity of the variance.

To facilitate the interpretation of the study across species, we enabled a direct visible and statistical comparison of the treatment effects through the normalization of each measured trait relative to its mean value for the outdoor treatment for each species. (Raw trait average values per species, treatments, and seasons are available in Appendix A). These data were used to perform one-way ANOVA, considering the treatments as fixed factor and species as random factors. As post hoc analysis, a Tukey pairwise multiple comparation test was used to identify significant differences among treatments.

Finally, to understand the variability of all the measured variables between treatments, principal component analysis (PCA) was performed separately for each species, using all measured traits as inputs. To complete the data set required for the PCA due to fewer gas exchange, pigment, and fluorescence measurements than the number of plants, in each species and treatment (*n* = 9), the missing values of chlorophyll content and light parameters were imputed using a normal distribution and keeping the same average and standard deviation of the performed measurements of the respective variables. All analyses were done using R [36].

## 5. Conclusions

The current study indicates that the implementation of fluctuations of temperature and light quantity are relevant at morphological and physiological levels to reach a more natural-like plant performance in indoor growth facilities. However, it became clear that although we were able to closely reproduce outdoor conditions in terms of temperature, humidity, and irradiance in our phytotrons, there were still significant differences prevailing in most investigated traits compared with field-grown plants. Other factors that could not be reproduced in our setup (e.g., soil temperature offsets, wind speed and directions) thus might have significantly added to the observed differences between indoor and outdoor grown plants. Although our results were generally species-specific, some general trends could be revealed. We recommend not using fixed day and night-time conditions, but to grow plants at least under sinusoidal climate fluctuations (as these are easier to apply and control than more random natural fluctuations) to, as shown in this study, reach more natural-like growth than under totally variable conditions if trait measurements on indoor plants are used for extrapolations to, and models of, natural systems. 

## Figures and Tables

**Figure 1 plants-09-01312-f001:**
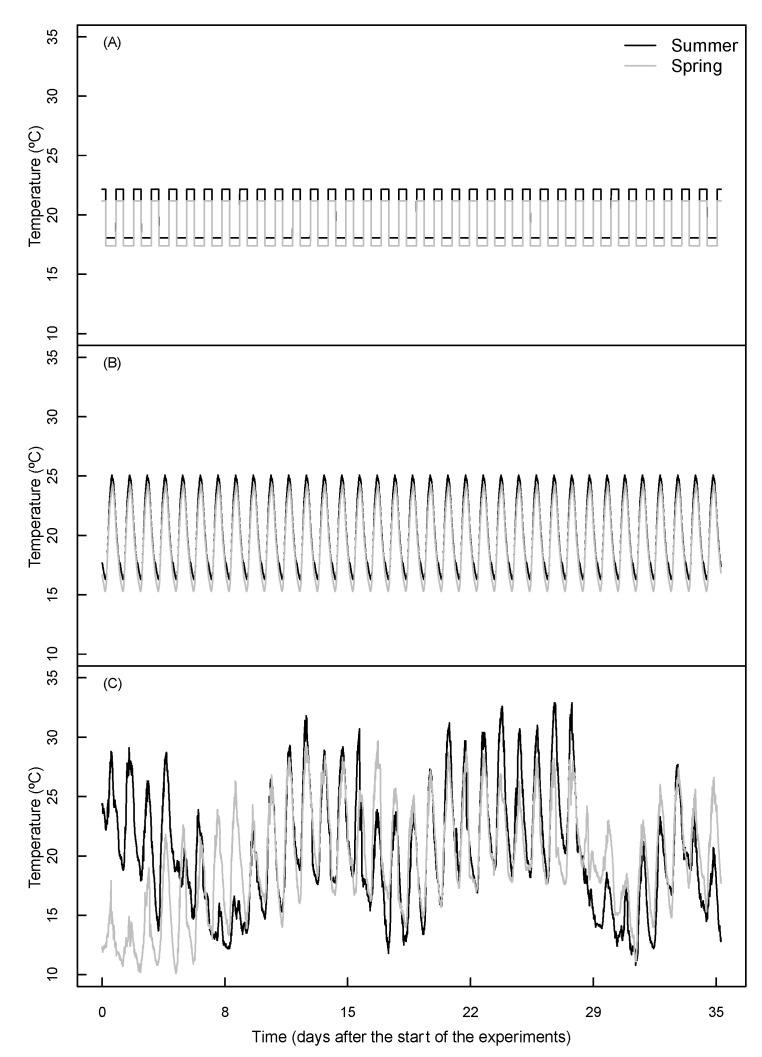
Applied temperatures for each treatment in the summer or spring run. Upper panel (**A**): fixed day and night conditions; middle panel (**B**): sinusoidal diurnal changes; lower panel (**C**): variable changes (real climate tracking). The corresponding relative humidity (%RH) and light quantity (PPFD as μmol m^−2^ s^−1^) conditions are available in the Appendix A.

**Figure 2 plants-09-01312-f002:**
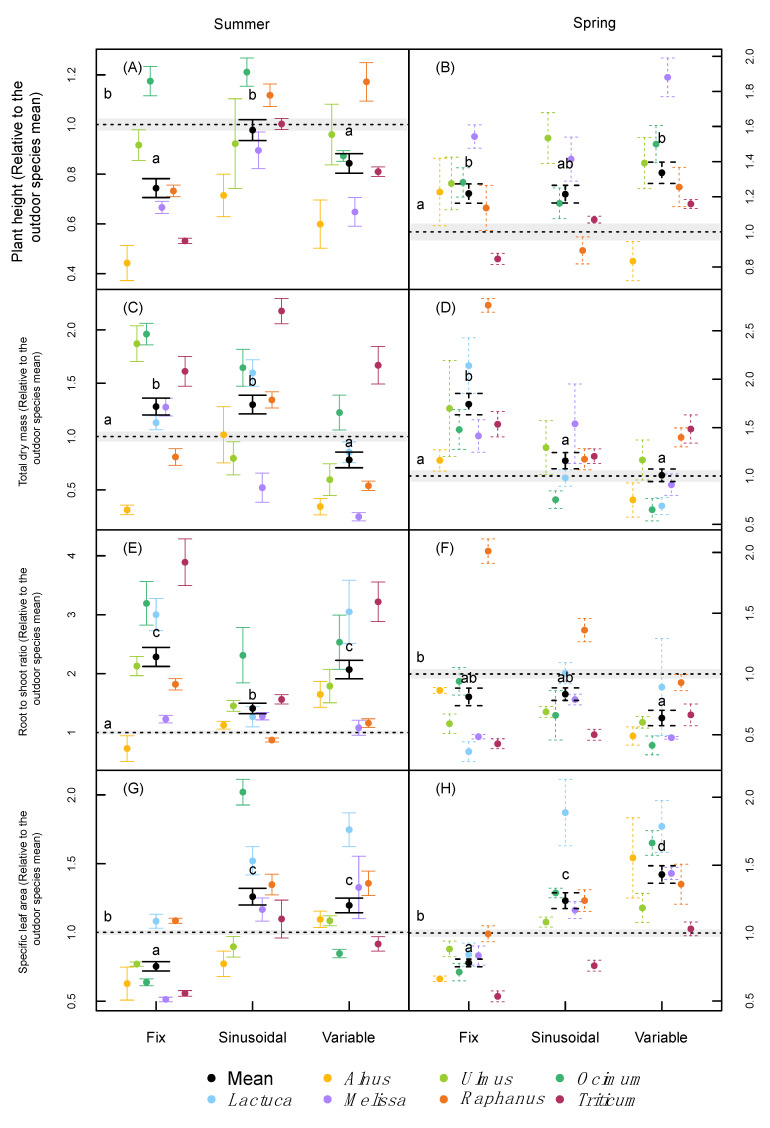
Plant height (**A**,**B**), total plant dry mass (**C**,**D**), root to shoot ratio (**E**,**F**), and specific leaf area (**G**,**H**) of the difference species normalized relative to the outdoor species mean under the two different runs (summer and spring). Error bars correspond to the standard error. *n* = 9 for each species. Letters indicate significant differences (*p* < 0.05, based on Tukey’s post-hoc tests) among treatments (including the outdoor trials) using species as a random effect, separately for each run. *Alnus* was not included in the spring trial under the sinusoidal treatment.

**Figure 3 plants-09-01312-f003:**
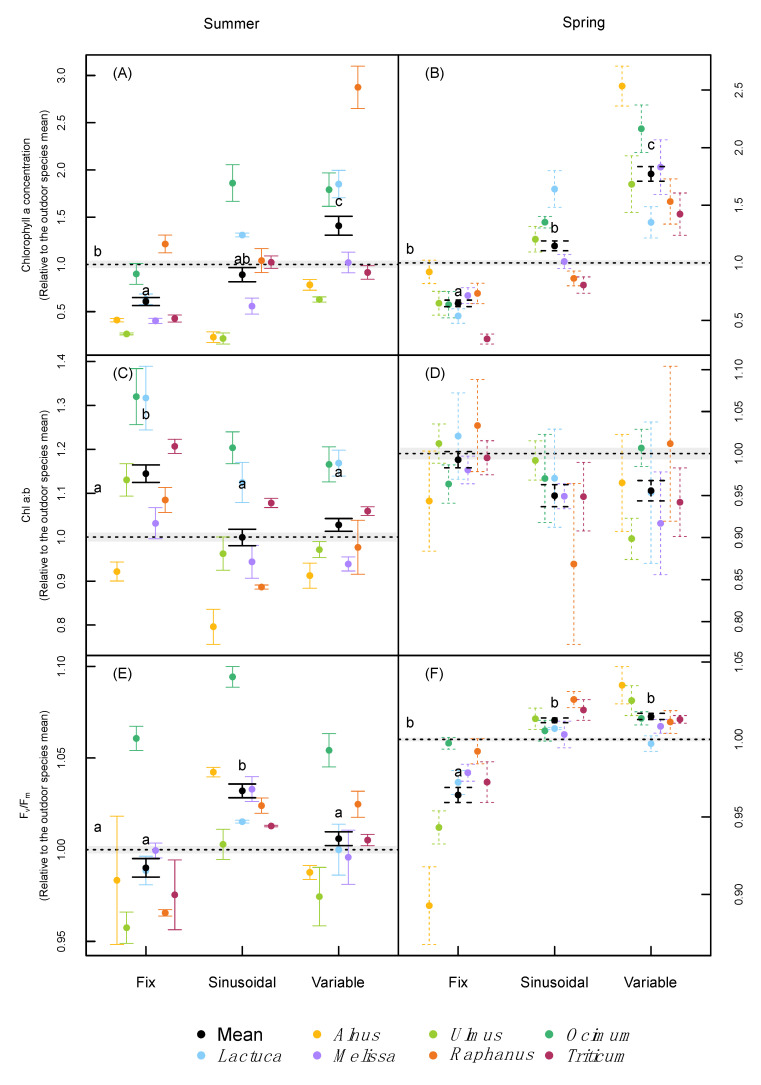
Chlorophyll a concentration (**A**,**B**), chlorophyll a to b ratio (**C**,**D**), and *Fv/Fm* values (**E**,**F**) of the difference species normalized relative to the outdoor species mean under the two different runs (summer and spring). Error bars correspond to the standard error. *n* = 4 for each species. Letters indicate significant differences (*p* < 0.05, based on Tukey’s post-hoc tests) among treatments (incl. the outdoor trials) using species as a random effect, separately for each run. *Alnus* was not included in the spring trial under the sinusoidal treatment.

**Figure 4 plants-09-01312-f004:**
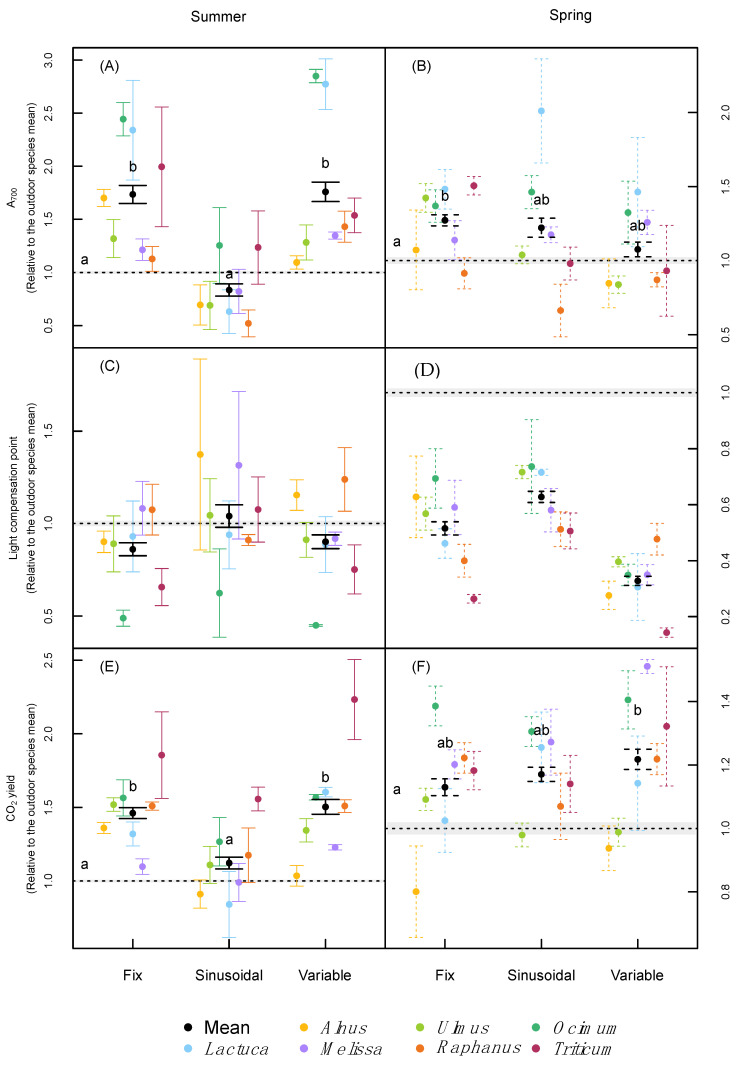
Maximum photosynthesis at 700 μmol m^−2^ s^−1^ PPFD (**A**,**B**), Light compensation point (**C**,**D**) and CO_2_ yield of photosynthesis (**E**,**F**) of the difference species normalized relative to the outdoor species mean under the two different runs (summer and spring). Error bars correspond to the standard error. *n* = 3 for each species. Letters indicate significant differences (*p* < 0.05 based on Tukey’s post-hoc tests) among treatments (incl. the outdoor trials) using species as a random effect, separately for each run.

**Figure 5 plants-09-01312-f005:**
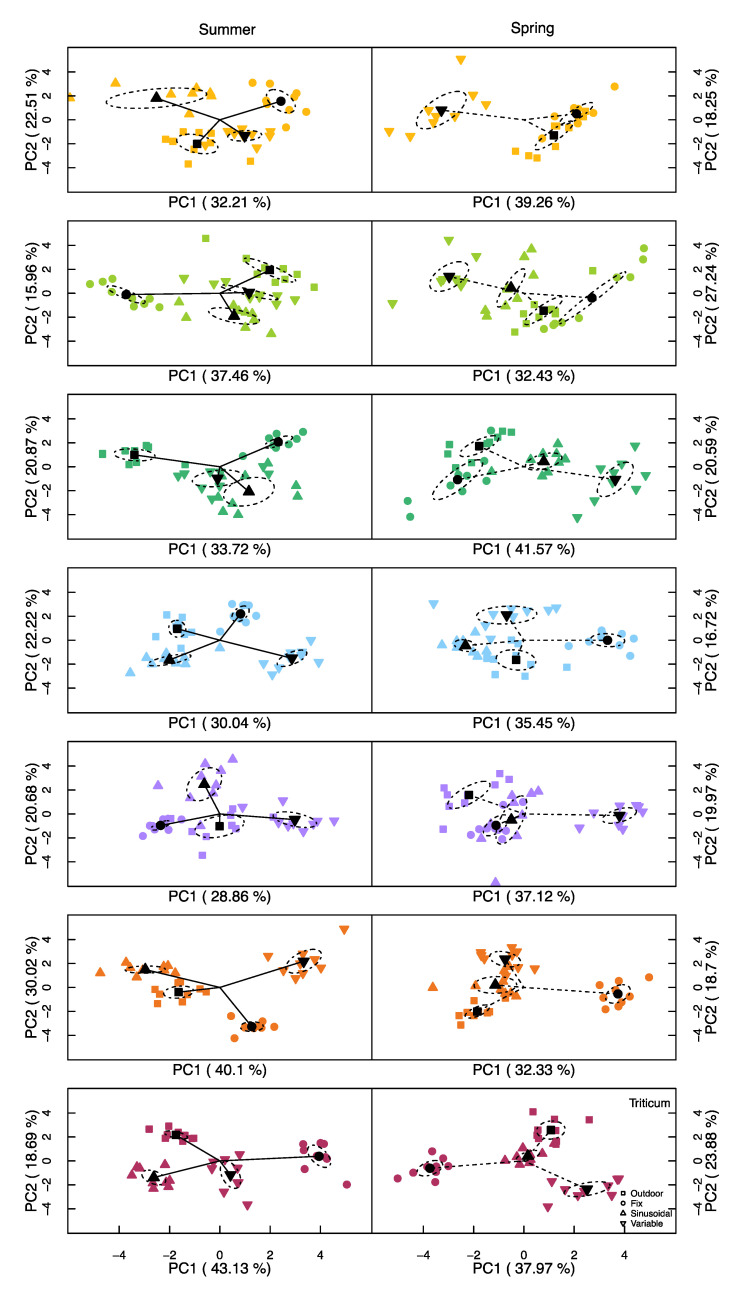
Principal component analysis of each species in the two runs (summer and spring). Ellipsoid calculated using the standard error of the plotted points. *Alnus* was not included in the spring trial under the sinusoidal treatment.

**Table 1 plants-09-01312-t001:** Average environmental conditions used for the summer and the spring runs in the phytotron experiments. The values are means across the respective 35-day growth periods of the field trials.

	Summer Run		Spring Run	
	Night	Day	Night	Day
Air temperature (°C)	18.1	22.2	17.4	21.2
Relative humidity (%)	79.2	64.9	81.7	67.4
PPFD (μmol m^−2^ s^−1^)	0	575.5	0	609
Duration per day (h)	11.05	12.95	9.92	14.08

**Table 2 plants-09-01312-t002:** *p*-values of the two-way ANOVA for all measured plant traits, separated for the summer and spring runs. Non-significant *p*-values (α = 0.05) are given in italics.

		Summer			Spring		
	Variable	Treatment	Species	Treatment × Species	Treatment	Species	Treatment × Species
Biomass and Morphology	Height **	<0.001	<0.001	<0.001	<0.001	<0.001	<0.001
	Dry weight shoot	<0.001	<0.001	<0.001	<0.001	<0.001	<0.001
	Dry weight roots	<0.001	<0.001	<0.001	<0.001	<0.001	<0.001
	Total dry weight	<0.001	<0.001	<0.001	<0.001	<0.001	<0.001
	Root to shoot ratio	<0.001	<0.001	<0.001	<0.001	<0.001	<0.001
	SLA	<0.001	<0.001	<0.001	<0.001	<0.001	<0.001
Chlorophyll	Chlorophyll α (mg g^−1^)	<0.001	<0.001	<0.001	<0.001	<0.001	<0.001
	Chlorophyll β (mg g^−1^)	<0.001	<0.001	<0.001	<0.001	<0.001	<0.001
	Chl α:β ratio	<0.001	<0.001	<0.001	*0.095*	<0.001	-
	Carotenoids (mg g^−1^)	<0.001	<0.001	<0.001	<0.001	<0.001	<0.001
	*Fv/Fm*	<0.001	<0.001	<0.001	<0.001	<0.001	<0.001
	Pi	<0.001	<0.001	<0.001	*0.445*	<0.001	-
Photosynthesis	Max. photosynthesis	<0.001	<0.001	-	*0.0567*	<0.001	-
	Light compensation point	*0.060*	<0.001	-	<0.001	0.005	-
	Quantum yield for CO_2_ fixation	<0.001	<0.001	0.006	0.003	<0.001	-
	Dark respiration	*0.5223*	<0.001	-	<0.001	<0.001	<0.001

** *Lactuca* was not included in this analysis. -: The variable was removed from the analysis due non statistically significant.

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
