# Peer review of "Reaching Natural Growth: The Significance of Light and Temperature Fluctuations in Plant Performance in Indoor Growth Facilities"

_plants, 2020, doi:10.3390/plants9101312_

Round 1

Reviewer 1 Report

The article presented to me for review raises very important issues related to the correct interpretation of research results when the researches are conducted indoor conditions, and the results of these experiments are recommended for field production. The manuscript presented for evaluation shows significant differences in the growth and development of various plant species depending on their growth conditions.
The results of the research show that the differentiation of changing climate conditions in the indoor gives more reliable research results, similar to those obtained in the conditions of natural growth and development of the studied plant species.

Details are provided in the manuscript

Author Response

Herewith, we are pleased to resubmit the revised version of our manuscript entitled "Reaching natural growth: The significance of light and temperature fluctuations on plant performance in indoor growth facilities” to the Plant journal after minor revisions

We have edited the details in the manuscript as recommended by the reviewers. We would like to thank the reviewers for their useful suggestions.

Camilo Chiang

Reviewer 2 Report

  • This is a very interesting and well written paper. I have really no comments. 

Author Response

(The authors gave the same response as above.)

Reviewer 3 Report

Review Chiang 2020 Plants

The question of how fluctuations in the environmental conditions affect plant growth and physiology is not new but has been increasingly investigated in recent years, partly due to advancements in glasshouse technology. Despite the many recent publications on the topic, the question is certainly interesting and of importance for basic and applied plant science. The present manuscript analyses the physiological parameters of eight plant species across the spectrum of functional and ecological groups, grown in different indoor climate regimes (termed “fixed”, “sinusoidal” and “variable”) in comparison with outdoor conditions. The experimental design is rather curious – the variable conditions are designed to exactly replicate the fluctuations in temperature, irradiance, and humidity of the outdoor experiment, in contrast to more typical climate chamber protocols. Essentially the authors tried to prove a trivial hypothesis (that plants grow similarly under similar conditions) that is almost true by definition – and failed to do so (the results were mostly species-dependent and, despite the large variations, in many cases the replicated variable climate caused more differences with the outdoor control than the sinusoidal regime). This could either be perceived as sloppy science (the lack of reproducibility) or as a provocative study highlighting the immense complexity of the outdoor environment and the difficulty in reproducing it simply by programming a small set of environmental factors, be it the ones most important for photosynthesis. I am inclined to favour the latter judgement and the Conclusions section accordingly hypothesizes that other factors (soil temperature, wind) might be the culprit for the observed variations (similar hypotheses have been raised in other studies as well). The dataset itself is solid and possibly merits publication; however, I would recommend the authors to focus on the novel findings and conclusions, make them clearer, and clarify the systematic analysis (in quantitative terms) that has led to the conclusions. If, indeed, fewer differences are found in the sinusoidal than in the variable indoor regime, what would be the possible reason?

Some of the main conclusions, e.g. the lower total biomass and higher specific leaf area under fluctuating climate conditions, are not novel and are well documented (in more studies than the few cited). The paper needs to show more clearly the key novel findings based on the experimental data presented. One might be that real-world-tracking variable phytotron conditions result in the most natural-like growth (204). However, as mentioned above, this is, one the hand trivial and expected and, paradoxically, not well demonstrated by the data. The definition of “natural-like” is lacking in the first place. It appears to be a qualitative label not assigned to any objective quantitative measure. Visually inspecting the mean differences in Figs 2-4, or the statistically different groups therein, or the distances in the PCA maps, one could arrive at the conclusion that the most “natural-like” are the plants grown under sinusoidal regime. The analysis must be elaborated and the way the authors arrive at the conclusions clarified.

The statement “we recommend to not use fixed day and night time conditions, but to grow plants at least under sinusoidal climate fluctuations” (489) could be elaborated (“at least” presumably stand for “or real-world-replicating fluctuations”; it is not clearly and explicitly stated why the sinusoidal regime is preferred, although there are hints to it in the figures).

The conclusion that “the effect of the diurnal temperature amplitude had a larger effect on height than the effect of fluctuating light” (229) is puzzling. The tautology notwithstanding (the effect had an effect), how were the effects of light and temperature variations differentiated from each other? It is not clear which experimental run had larger variations in temperature (but not light) and which in light (but not temperature).

Further/minor points

  • The plant species are not introduced. Apparently, this is because the authors intended the Materials section to precede the results. The Materials section contains the phrase: “In the following the species will always be referred to by their scientific gene name.” The section, however, is by the end of the manuscript.
  • The terms “fixed” and “sinusoidal” are somewhat inaccurate and could be misleading. In the “fixed” treatment, the temperature, irradiance and humidity are varied following a square wave (granted, they can be considered fixed during the day and night time). In the “sinusoidal” treatment the parameters are continuously altered following a periodic function, which is not sinusoidal, according to Fig. 1 and supplementary Fig. 1. Some clarification would be useful.
  • Species should be indicated in Fig. 5.
  • 205: “phytotron runs that simulated the real temperature and humidity variations” implies that light had less of an impact on the plant characteristics.
  • 480: “implantation” – probably meant “implementation”

Author Response

(The authors gave the same response as above.)
